# Malaria Vectors and Vector Surveillance in Limpopo Province (South Africa): 1927 to 2018

**DOI:** 10.3390/ijerph17114125

**Published:** 2020-06-09

**Authors:** Leo Braack, Riana Bornman, Taneshka Kruger, Yael Dahan-Moss, Allison Gilbert, Maria Kaiser, Shüné V. Oliver, Anthony J. Cornel, Yoosook Lee, Douglas E. Norris, Maureen Coetzee, Basil Brooke, Christiaan de Jager

**Affiliations:** 1University of Pretoria Institute for Sustainable Malaria Control, School of Health Systems and Public Health, Faculty of Health Sciences, University of Pretoria, Pretoria 0028, South Africa; riana.bornman@up.ac.za (R.B.); taneshka.kruger@up.ac.za (T.K.); ajcornel@ucanr.edu (A.J.C.); tiaan.dejager@up.ac.za (C.d.J.); 2Vector Control Reference Laboratory, Centre for Emerging, Zoonotic & Parasitic Diseases, National Institute for Communicable Diseases, Johannesburg2001, South Africa; Yaeld@nicd.ac.za (Y.D.-M.); allisong@nicd.ac.za (A.G.); mariak@nicd.ac.za (M.K.); shuneo@nicd.ac.za (S.V.O.); maureen.coetzee@wits.ac.za (M.C.); basilb@nicd.ac.za (B.B.); 3Wits Research Institute for Malaria, School of Pathology, Faculty of Health Sciences, University of the Witwatersrand, Johannesburg 2001, South Africa; 4Department of Entomology & Nematology, University of California, Davis, CA 95616, USA; yoosook.lee@gmail.com; 5The W Harry Feinstone Department of Molecular Microbiology and Immunology, Johns Hopkins Malaria Research Institute, Johns Hopkins Bloomberg School of Public Health, Baltimore, MD 21205, USA; douglas.norris@jhu.edu

**Keywords:** malaria, *Anopheles*, Limpopo Province, South Africa, vector surveillance

## Abstract

Despite the annual implementation of a robust and extensive indoor residual spraying programme against malaria vectors in Limpopo Province (South Africa), significant transmission continues and is a serious impediment to South Africa’s malaria elimination objectives. In order to gain a better understanding regarding possible causes of this residual malaria, we conducted a literature review of the historical species composition and abundance of malaria vector mosquitoes in the Limpopo River Valley region of the Vhembe District, northern Limpopo Province, the region with the highest remaining annual malaria cases in South Africa. In addition, mosquito surveys were carried out in the same region between October 2017 and October 2018. A total of 2225 adult mosquitoes were collected using CO_2_-baited tent and light traps, human landing catches and cow-baited traps. Of the 1443 *Anopheles* collected, 516 were members of the *An. gambiae* complex and 511 *An. funestus* group. In the malaria endemic rural areas outside the Kruger National Park, one specimen each of *An. gambiae s.s.* and *An. funestus* and only three of *An. arabiensis* were collected. The latter species was abundant at a remote hot spring in the neighboring Kruger National Park. Eighteen other species of *Anopheles* were collected. Our survey results support the historical findings that *An. arabiensis*, the species widely held to be the prime malaria vector in South Africa, is a rare species in the malaria endemic Limpopo River Valley. The implications of the mosquito surveys for malaria transmission, elimination and vector control in northern Limpopo Province and neighboring regions are discussed.

## 1. Introduction

The more northern regions of the Republic of South Africa have historically suffered heavily from the impact of malaria, reflected in both popular and scientific accounts [1,2,3,4]. Currently, South Africa has three malaria endemic provinces with sustained annual local transmission—Limpopo, Mpumalanga and KwaZulu/Natal. Limpopo Province, which borders Zimbabwe to the north and Mozambique to the east, has the highest malaria incidence [3,4].

It was the discovery in 1898 that *Anopheles* mosquitoes are responsible for malaria parasite transmission [5], that spurred efforts to incriminate the specific local species responsible for primary transmission around the world. In South Africa, it was only in the mid 1920’s that extensive initial surveys of mosquitoes were carried out in the previous Transvaal and Natal Provinces [6,7,8]. At the time nothing was known about the complexities of sibling species and the variable vectorial capacities within such species complexes. The work of Paterson [9,10], Davidson [11], Coluzzi [12], and others, using time-consuming cross-mating studies and chromosomal methods, did much to elucidate and unravel the species relationships in the 1960’s and 70’s.

Historically, for the first half of the 20th century prior to the introduction of regular annual indoor spraying of insecticides, *Anopheles gambiae* and *Anopheles funestus* were the main vectors of malaria in South Africa [2,6,7,8], but the concerted indoor control efforts resulted in local extinction of both species. Since then it has been widely believed that *Anopheles arabiensis* is the primary remaining vector responsible for malaria transmission in South Africa.

The malaria vector control programmes in the three endemic Provinces undertake annual application of indoor residual spraying (IRS) using a mosaic approach comprising two different classes of insecticides, pyrethroids and organochlorines. They also conduct vector surveillance which for decades has been limited largely to the *An. gambiae* complex, mainly by way of larval collections. Other Provinces, such as Northwest and Gauteng, experience significant levels of imported malaria cases but no local transmission.

Initially, the South African vector surveys included larval collections as well as human landing catches and indoor pyrethrum knockdowns, but in more recent years surveillance has focused on larval collections supplemented to some extent by adult pit-resting and container-resting (pots and jars) collections. Some provinces have the capacity to undertake their own molecular identification of the species complexes but support for identifications is provided largely by the National Institute for Communicable Diseases in Johannesburg.

Limpopo Province presents a pernicious residual malaria challenge for which the causal reasons remain unknown, especially as annual IRS is rigorously applied and is combined with active surveillance to detect infection sources and rapid treatment of all passive and actively diagnosed cases. This paper reports on a literature review of the historic species composition of *Anopheles* mosquitoes in Limpopo Province, and on current species composition and abundance based on independent mosquito collections carried out in 2017/18 in the far north-eastern region of the Province. We discuss the implications of these findings on malaria transmission in the area 

## 2. Materials and Methods

### 2.1. Study Area

This study took place in the malaria endemic region of northern South Africa in what is currently known as Limpopo Province, one of four sub-divisions of what was previously the much larger “Transvaal” Province. Historically, malaria was present over a significantly larger area but sustained malaria control interventions dating back nearly 90 years has shrunk this malaria endemic region considerably [4]. The general Limpopo Province region was the focus for the literature review.

Within Limpopo Province, the Vhembe Municipal District is the area most heavily impacted by malaria in South Africa. It is located in the extreme north-eastern region of South Africa directly adjoining Zimbabwe and Mozambique, countries with which there is substantial trans-border movement of people, such as temporary migrant workers. The low-lying Limpopo River valley within this north-eastern corner has a high annual malaria incidence (in the context of South Africa), and we conducted mosquito surveillance in or near rural settlements or settings known for high malaria risk, plus random catches in the general Limpopo River valley area for comparative purposes (Figure 1). The sites we used for core sampling were Bende Mutale (22°25.117′ S, 31°02.016′ E), Popallin Ranch (22°21.248′ S, 30°35.434′ E) and the worker village at Doreen Farms (22°30.277′ S, 30°14.119′ E). We also report findings of opportunistic comparative catches at Tshipise tourist resort (22°36.360′ S, 30°10.403′ E), and in the Kruger National Park at Pafuri (Crook’s Corner) (22°26.445′ S, 31°18.719′ E) and Malahlapanga hot spring (22°53.374′ S, 31°02.391′ E).

### 2.2. Data Sources

We conducted a literature search using Google Scholar and PubMed. Keywords used were “Malaria AND South Africa” as well as “Malaria AND vectors AND South Africa”. Timeframe was not specified, therefore accessing all records ever published, at least those accessible by the two search platforms used. Titles and abstracts of the search finds were then examined for relevance, and appropriate ones examined further. The most useful of these papers relevant to this study are summarized in Table 1.

### 2.3. Vector Surveillance and Methods Used for New Data Presented Here

Vector surveillance was targeted at known high-incidence and recurring malaria “hotspots” in the eastern Limpopo River valley region of Vhembe District. Collections were made in October and December 2017 and again in February, April and October 2018. Adult mosquitoes were collected using CO_2_-baited light traps and tent traps baited with dry ice CO_2_, cows and sometimes goats (Figure 2), and occasionally human landing catches. Ethics approval for human landing catches and for use of cattle as bait in tent traps were obtained from the University of Pretoria (code H015-17). In addition, larval collections were made from a wide range of pools along river edges or other forms of surface water, using the standard dip method or surface sweep-netting [18]. In most cases only *Anopheles* mosquitoes were identified to species level and other culicines recorded at genus level, although when time and expertise was available these other non-anophelines were also identified to species. *Anopheles* specimens were all—except for unidentifiable damaged specimens—microscopically identified to species or species group/complex within hours after capture, using standard reference keys [19]. All anophelines were placed individually in silica-gel reaction tubes for Polymerase Chain Reaction (PCR) species identification of members of the *An. funestus* group [20] and the *An. gambiae* complex [21], and subsequent testing for sporozoites [22].

## 3. Results

The publications providing insight and understanding of malaria vectors and associated mosquito communities over almost a full century of research and surveillance in what is now the Limpopo Province of South Africa, are sufficiently few that the findings in each can be summarized below.

### 3.1. Historical Surveys in Limpopo Province

The first records reflecting *Anopheles* species composition and some idea of distribution and general abundance are those of Ingram and De Meillon [6,7], published in 1927 and 1929. They report on mosquito surveys conducted along the railway lines in the “northern Transvaal” (Limpopo Province) and “eastern Transvaal” (Mpumalanga Province), and along the coastal belt of Zululand (the northern regions of KwaZulu-Natal Province). These publications are the first detailed surveys with published reports of mosquitoes in what is now Limpopo Province, no significant collections having been undertaken prior to that, as corroborated by these authors. In their coverage of the “northern Transvaal” in 1926 and 1928, they recorded a total of 16 anopheline species (Table 2). Their records of “*An. funestus*” and “*An. gambiae*” would have included some of the other species now known to occur within these groups/complexes, and in January 1928 they recorded *“An. gambiae* var. *quadriannulatus*” from Leydsdorp (near Tzaneen, Figure 1). Based largely on the known status of these two malaria vector groups as primary vectors elsewhere in Africa, they suggested that these were likely to be the main vectors in South Africa, despite the curious anomaly of long periods of undetectable presence of either species in areas experiencing malaria transmission, in the case of “*An. gambiae*” even for 3 or 4 years in succession. That anomaly remains relevant today.

Despite being limited in scope and duration, the survey by Swellengrebel et al. [2] yielding 877 anophelines in the “Transvaal” to determine parasite presence in *Anopheles* species, does present useful information, summarized in Table 3. They discuss six species of *Anopheles* collected mostly from inside human dwellings, of which *Plasmodium* parasites were recovered commonly from *An. funestus* and *An. gambiae* in two different locations and also from *An. pretoriensis* collected both inside and outside houses in the Letaba foothills outside Tzaneen (Figure 1). These results also confirmed the distribution of the two vector groups, with *An. funestus* being restricted to the foothills along the mountain range and members of the *An. gambiae* complex more prevalent in the lowland areas. Later studies provided evidence for the restricted distribution of *An. funestus* being due to its preferred larval habitat of small streams or rivers in the area [8].

Steyn et al. [13] report on a total of 538 mainly culicine mosquitoes collected during a March 1953 survey along the upper Limpopo River Valley between Vaalwater and Pafuri (Figure 1). The focus of the survey was to understand culicine mosquito composition in the area, and the emphasis was on larval collections, especially tree-hole breeding sites. Three genera comprising 21 species of mosquitoes were collected, including six species of *Anopheles*, nine of *Aedes* and six *Culex* (Table 4). This is the only historical survey recording the presence of *An. listeri*, collected to the east of Musina close to where the current survey was conducted.

The paper by La Grange and Coetzee [14] is especially useful as it reports on a detailed list of anopheline species collected using four different methods over a 27-month June 1987–August 1989 period in the village of Thomo, Limpopo Province (near Giyani Figure 1). Direct comparisons of anopheline species composition and relative abundance can therefore be made with the October 2017–October 2018 survey conducted in Vhembe District as reported below. Catch details for the 1987/89 collections are provided in Table 5. Of the 23,252 anophelines collected, 85.8% comprised members of the *An. funestus* group (no identifications of specific species was done), while members of the *An. gambiae* complex made up a mere 1.05% (*n* = 245, of which a sample of 155 were identified electrophoretically [23] as *An. quadriannulatus*. Of the 2994 anophelines (11 species) caught by human landing catches, 70.1% were *An. coustani* and 28.1% *An. funestus* group, with only six individuals belonging to the *An. gambiae* complex (0.2%). No *An. arabiensis* were identified in the 155 processed for electrophoretic identification.

The northern half of the roughly 19,000 km^2^ Kruger National Park (KNP) forms part of Limpopo Province and is located as a pristine, untransformed wedge of land between Mozambique and the rest of Limpopo Province. This northern part of the KNP has been the target of several mosquito surveys in recent decades, some focused on the unique freshwater spring Malahlapanga which offers ideal breeding conditions for select members of the *An. gambiae* complex [16,24,25,26], and also along the Shingwedzi River in very different dense woodland settings [17]. The collections from these various surveys offer interesting comparative insights with collections in more transformed settings elsewhere in Limpopo Province. At the Malahlapanga freshwater spring, with water slowly bubbling out at 37 °C and forming numerous footprint pools in the shallow overflow stream where animals come to drink, several publications [16,24,26] report on the strong presence of *An. arabiensis* since the late 1980s, to the near total exclusion of other members *An. gambiae* complex. This state of *An. arabiensis* dominance continued until about 2012, as reflected in surveys by Munhenga et al. [16] between July 2010 and December 2012 (Table 6), but for unknown reasons the population then rapidly changed to being dominated by *An. quadriannulatus* (Braack, Munhenga, personal observations). Nevertheless, for purposes of this paper, the findings of the Munhenga et al. 2014 survey [16] serve as a useful reference point and are summarized in Table 6. It should be pointed out that very distinct differences existed in the *Anopheles* communities associated with each of these five sites.

With the exception of Malahlapanga with its unique attributes affording an ideal breeding opportunity for *An. arabiensis* (99.6% of the 1352 specimens collected), this species was very rare at all the other sites. Similarly, 98.6% of 348 *An. merus* were caught at the two salt-water springs Mafayeni and Matiovila, in line with their larval biology. Also interesting, despite the relatively isolated nature of Malahlapanga, having no connection with nearby watercourses for much of the year and its setting dominated by monotonous mopane-woodland, this site had the highest diversity of anopheline species (all nine species were found here). It also had the highest numbers (75.4% of overall total mosquitoes captured at all sites) whereas the site “Louis-se-Gat” adjoining the Shingwedzi River, with diverse and lush riverine forest frequented by an abundance of diverse mammals and birds, yielded only 4.6% of the total catch. 

Cornel et al. [17] at Shingwedzi in this northern part of the Kruger National Park, collected 168 mosquitoes comprising 20 species in five genera during three nights of collections using CO_2_-baited net traps and light traps. Of the 20 species, eight were *Anopheles* (Table 7) accounting for 65% of the total catch. Cornel et al. also sampled mosquitoes in the Lapalala Nature Reserve in the western interior region of Limpopo Province, using CO_2_-baited net traps and light traps over four nights. Here they collected a total of 296 mosquitoes comprising 19 species in five genera. Of these, five species were *Anopheles*, comprising 49% of all mosquitoes captured (Table 7). No *An. arabiensis* was recorded from either site.

### 3.2. Surveys in Surrounding Areas

For comparative purposes, it is useful to have a sense of *Anopheles* community composition in the neighboring malaria regions of Mpumalanga Province to the south, Zimbabwe to the north, and Mozambique to the east.

In Mpumalanga Province, Govere et al. [15] conducted monthly collections of mosquitoes at 7 sites in the Lowveld Region between August 1997 and May 1998 (Table 8). A total of 5084 *Anopheles* were collected of which *An. coustani* was by far the most abundant (*n* = 2837, 55.8%), followed by the *An. funestus* group (*n* = 1418, 27.9%, species not identified) and the *An. gambiae* complex (*n* = 435, 8.6%). *Anopheles pretoriensis* made up 5.2% of the total catch while the remaining 2.6% comprised of *An. demeilloni*, *An. longipalpis*, *An. maculipalpis*, *An. marshallii*, *An. rufipes* and *An. squamosus*. Members of the *An. gambiae* complex were *An. merus* (56%), *An. quadriannulatus* (30.4%) and *An. arabiensis* (13.6%). However, more than 80% of the *An. gambiae* complex catch came from only one (Martiens) of the seven collection sites, thus distorting the interpretation of results, in particular the abundance of *An. merus*. ELISA assays for *Plasmodium falciparum* circumsporozoite antigen presence were negative for all *An. gambiae* complex members.

The high percentage of *An. merus* relative to other members of the *An. gambiae* complex reported in at least one area of Mpumalanga Province [15] was subsequently substantiated by a similar study undertaken by Mbokazi et al. [27], who found steadily increasing abundance of *An. merus* and greatly expanded distribution across Mpumalanga Province over a nine-year period, 2005 to 2014.

North of Vhembe District, Sande et al. [28] conducted anopheline surveys in the Mutare and Mutasa Districts of Manicaland Province, Zimbabwe, during the period November 2013 to April 2014. They sampled larvae in a range of habitat types and conducted indoor pyrethrum knockdown spray catches for adult anophelines each month. Approximately 4848 *Anopheles* larvae were collected yielding 4690 adults, of which 97.9% (*n* = 4593) were *An. pretoriensis*, while *An. funestus* group members comprised 1.9% (*n* = 87) and *An. gambiae* complex members 0.2% (*n* = 10).

Cumulatively, considering all anophelines reared from larval collections and adults captured by way of knockdown catches indoors, members of the *An. funestus* group were 27 times more abundant than the *An. gambiae* complex. From a total of 840 *An. funestus* group females subjected to PCR assay [20], 90.8% were *An funestus*, 5.1% *An. leesoni* and the rest failed to amplify. Of 31 *An. gambiae* complex females assayed by PCR [21], 48.4% (*n* = 15) were *An. quadriannulatus*, 41.9% *An. arabiensis* (*n* = 13) and 9.7% were unidentifiable.

In summary, *An. arabiensis* was present only in very low numbers over the entire period, with *An. funestus* being relatively more abundant but in terms of overall numbers also very low, especially in comparison with *An. pretoriensis*. A summary of mosquito catches is provided in Table 9. These findings confirm earlier reports of Masendu et al. [29] that showed dominance of *An. pretoriensis* over other anopheline species in Zimbabwe. However, the findings also contrasted with previous reports [30,31] that *An. arabiensis* was the main vector in these regions, and instead indicated that unexplained population changes had occurred which resulted in *An. funestus* becoming more common and taking over as the probable main malaria vector, especially given its strong endophilic and anthropophagic habits.

Worth noting are the recent findings of Zengenene et al. [32], who report on limited collections of *Anopheles* adults and larvae from the Chiredzi District in Zimbabwe, directly adjacent to our Limpopo Valley study area and at collection sites located a relatively short distance of approximately 130–150 km from our primary sampling areas. From a total of 153 *Anopheles* collected, no specimens of *An. arabiensis* or *An. merus* were found, although *An. quadriannulatus* was common. Of the 16 members of the *An. funestus* group collected as adults, 14 (87.5%) were *An. funestus*. Despite the small sample size, these findings are significant and are discussed below.

In neighboring southern Mozambique, Casimiro et al. [33] and Kyalo et al. [34] report on the presence of *An. gambiae s.s.* while *An. funestus* has been recorded by Brooke et al. [35] and Casimiro et al. [36,37].

### 3.3. 2017/2018 Vector Surveys

Surveillance activities in the Limpopo Valley region of Vhembe District between October 2017 and October 2018 yielded a total of 2225 adult mosquitoes representing 8 genera, of which *Anopheles* constituted 64.9% (*n* = 1443) of the total. *Culex* (17.3%), *Aedes* (13.2%) and *Mansonia* (4.4%) made up most of the remaining numbers with a few specimens of *Mimomyia*, *Coquellittidia* and *Aedeomyia* also captured (Figure 3). Of the 1443 *Anopheles* captured, 1027 (71.2%) belonged to four members of the *An. gambiae* complex (35.8%) and six members of the *An. funestus* group (35.4%). Eleven other species were collected with *An. pretoriensis* (9%), *An. rufipes* (5.8%) and *An. listeri* (3.5%) being the most common (Figure 4). The relatively small sample (*n* = 391) of adults reared from larval collections yielded a species abundance that largely reflects that of the adult collections (Figure 4), except for the *An. funestus* group, which featured very poorly in the larval collections reflecting the well-known difficulty of finding these larvae at low densities ([38], p. 134).

Of the 511 *An. funestus* group collected, 408 individuals were successfully assayed by PCR [20] for species separation (Table 10). Considered collectively across all sites and all sampling months, *An. rivulorum* formed the overwhelming majority (*n* = 312). 

A single specimen of *An. funestus* was collected at the Popallin Ranch tourist lodge in February 2018. This was somewhat unexpected as *An. funestus* is generally regarded to have been eliminated from Limpopo Province for decades due to regular and widespread application of indoor residual application of Dichlorodiphenyltrichloroethane (DDT) in this region. However, its identity is confirmed and *An. funestus* is known to occur commonly in neighboring Zimbabwe at sampling sites approximately 130 km away [32]. While *An. rivulorum* was the dominant species of the group across most sites, at the Tshipise tourist resort *An. vaneedeni* comprised 88% in April 2018 (Table 10). These were collected from two CO_2_-baited Centres for Disease Control and Prevention (CDC) light traps placed alongside a wetland overgrown with a dense stand of reeds (*Phragmites australis* (Cav.) Trin. ex Steud.), adjoining the camping area.

Of the 516 *An. gambiae* complex collected, 207 specimens were assayed by PCR [21]. Of these, 150 (72.5%) were *An. quadriannulatus*, 48 (23.2%) *An. arabiensis* and 8 (3.9%) *An. merus* (Table 10). Of the 48 *An. arabiensis* collected over the one-year period, 42 were collected during a single night of sampling in October 2018 at the Malahlapanga freshwater spring in the Kruger National Park. One single specimen of PCR-determined *An. gambiae s.s.* was collected at Popallin Ranch in February 2018. *Anopheles gambiae s.s.* is known to have occurred historically widely across southern Africa [25], including the northern parts of South Africa [39,40].

Figure 5 depicts the relative abundance of *Anopheles* species collected at the three core sampling areas of Doreen Farms, Popallin Ranch, and Bende Mutale and at the opportunistic sampling sites of Tshipise tourist lodge, Pafuri, and Malahlapanga. The percentage contribution of the *An. funestus* group and *An. gambiae* complex for five sites (excluding Malahlapanga) is shown in Figure 6.

### 3.4. Sporozoite Assays

A sample of 369 anopheline mosquitoes were tested using a *Plasmodium* sporozoite multiplex PCR assay [22]. Samples included 125 *An. rivulorum*, 18 *An. rivulorum*-like, 46 *An. vaneedeni*, 13 *An. leesoni*, 82 *An. quadriannulatus*, seven *An. merus*, three *An. arabiensis* and 75 specimens of various *Anopheles* species. All were negative. This lack of finding sporozoites may simply reflect inadequate sample sizes for some of the vector species.

## 4. Discussion

The epidemiological picture has changed dramatically in the malaria-endemic regions of South Africa, from the historic setting of the early 20th century when it became known that *Anopheles* mosquitoes were the main agents of transmission, to the present. Poverty continues to remain widespread in rural areas and large numbers of people still live in sub-optimal housing offering inadequate protection against entry by mosquitoes. A lack of electricity with inability to afford propane gas necessitates outdoor cooking with charcoal or wood, and inadequate ventilation drives people outdoors to cool down in the evening hours. However, almost a century of State-implemented vector control has resulted in a shift in malaria parasite transmission dynamics. South Africans were global pioneers in testing and implementing indoor house spraying against what were the two primary vectors, *An. gambiae* and *An. funestus*, in the first half of the 20th century, demonstrating in the early 1930s that indoor spraying resulted in dramatic reduction of mosquitoes and malaria [8,41,42]. This led to the adoption in South Africa (and very soon also multiple other countries in the world) of house spraying as a key component of a formalized vector control strategy, at first using a 1:18 mixture of pyrethrum/paraffin [41,42] for spraying on walls but changing to dichlorodiphenyltrichloroethane (DDT) in 1946, the latter remaining the primary vector control intervention in South Africa until 1996, when pyrethroids were added due to public concern about DDT toxicity [3,4,43,44]. While this led to the disappearance of *An. gambiae s.s.* and *An. funestus* from the country, the potential for very rapid re-invasion and resurgence from neighboring countries was amply demonstrated in the late 1990s when, unfortunately, *An. funestus* in southern Mozambique had developed insecticide resistance to pyrethroids [45]. When DDT was removed, this vector came back into South Africa causing a major malaria epidemic [3,4]. It is widely believed that as a consequence of IRS suppressing the major indoor-resting vectors, the outdoor-biting *An. arabiensis* has for a long time been the primary vector of malaria in large parts of South Africa [46,47,48,49]. This conviction is likely to hold true in KwaZulu-Natal where *An. arabiensis* is common and known to harbour *Plasmodium* parasites [49], but some scepticism may be appropriate for the more northern areas of South Africa where the species is consistently recorded at very low levels [19,26].

### 4.1. Vector Status of Members of the Anopheles Gambiae Complex and Their Role in the Limpopo River Valley

In KwaZulu-Natal Province, *An. arabiensis* is common and has recently been implicated in malaria transmission [49], but various studies suggest that it is a rare species further north, especially in Limpopo Province [19,26], including our own findings during the 2017–2018 survey of the Limpopo River Valley. Two other members of the *An. gambiae* complex occur widely in the warm, lowland regions of north-eastern South Africa where malaria remains endemic. *Anopheles quadriannulatus* is known to host *P. falciparum* in laboratory studies [50] but has thus far never been incriminated as a vector in nature. This species has an interesting and apparently fluctuating presence across its area of distribution in South Africa, known to be abundant in Limpopo and Mpumalanga Provinces [15] but more uncommon in KwaZulu-Natal. It’s presence at Malahlapanga in northern Kruger National Park has raised considerable interest in what drives population abundance in a setting that offers apparently perfect breeding conditions year-round. In this particular wildlife setting *An. arabiensis* maintained complete dominance for several decades to the near total exclusion of *An. quadriannulatus* [16,24,26] but the situation was inexplicably reversed sometime around 2012 (Braack, Munhenga, personal observations), with *An. quadriannulatus* dominating the *Anopheles* community despite no detectable difference in breeding site availability, water composition or quality, or wildlife host composition or abundance. Our October 2018 sampling (Table 10) suggests that *An. arabiensis* has re-established its previous dominance at the site. Such shifts highlight our poor understanding of the factors affecting population distribution and abundance. However, *An. quadriannulatus* only reluctantly feeds on humans so that despite its known ability to host *P. falciparum* it is unlikely to be a significant factor in overall malaria epidemiology. Our survey in the Limpopo River Valley showed that *An. quadriannulatus* is common in inhabited rural areas but *An. arabiensis* is rare (aside from Pafuri and Malahlapanga in the wildlife conservation area of the Kruger National Park with few people at these sites). It is difficult to reconcile the generally accepted non-vector status of *An. quadriannulatus* and the scarcity of *An. arabiensis* with the rather high seasonal transmission of malaria that occurs in this northern region of South Africa, which suggests other vectors play a more important role.

*Anopheles merus*, a saltwater-breeding member of the *An. gambiae* complex, is widely distributed across the malaria-endemic provinces of South Africa, generally at low levels except in specific areas where it becomes locally abundant. This species too displays apparent adaptive plasticity, being able to shift from its more usual association with brackish-water breeding sites [38,51] in coastal areas, to exploit both saltwater and freshwater environments much further inland, such as in South Africa [25]. *Anopheles merus* is broadly distributed in the eastern and south-eastern regions of sub-Saharan Africa and adjacent Indian Ocean islands [52] and is an important vector of malaria in many areas where it occurs [51,53,54]. However, its very low presence in the malaria-afflicted regions of the Limpopo River Valley, as reflected in our survey results summarized in Table 10, suggest that it is not an important contributor to malaria transmission in this area.

The single specimen of *An. gambiae s.s.* collected in February 2018 at Popallin Ranch on the Zimbabwe border requires further investigation. Earlier records show the presence of this species at Sibasa, Thohoyandou, Limpopo Province in 1974 [39] and Pelindaba, northern KwaZulu-Natal Province in 1977 [40], as well as neighbouring Zimbabwe [23] and Mozambique [33].

### 4.2. Vector Status of Members of the Anopheles Funestus Group and Their Role in the Limpopo River Valley

Various members of the *An. funestus* group have been recorded in Limpopo Province [14,19,43]. Apart from *An. funestus*, all other species in the group have been regarded as non-vectors until recently when Burke et al. [55] confirmed the presence of *P. falciparum* sporozoites in *An. vaneedeni* in Mpumalanga and KwaZulu-Natal Provinces. Four species, *An. leesoni*, *An. parensis*, *An. rivulorum* and *An. rivulorum*-like, were recorded from Shingwedzi in the Kruger National Park in 2015 [17,56]. In this current paper we report a high abundance of *An. rivulorum* in the Vhembe District of Limpopo Province, together with several other members of the *An. funestus* group (Table 10). Of these, *An. rivulorum*, *An. leesoni*, *An. parensis* and *An. vaneedeni* are known to host *P. falciparum* [48,55,57,58,59], thus demonstrating the potential of these species as vectors in residual malaria settings. We suggest that given the known ability of members of the *An. funestus* group to host *P. falciparum*, these species should be considered as likely to be significant vectors of malaria in this region. Even if only a small proportion of specimens are infected, their collective contribution would make them significant.

Based on human landing catches in Limpopo Province, La Grange and Coetzee [14] reported *An. funestus* group females as being most active in the first two hours after sunset, which together with the known status of anthropophagy, exophagy and sporozoite positivity [59], predisposes at least some of these species as potential outdoor secondary vectors. Furthermore, these species are not influenced to any great extent by rainfall [8], relying on year-round breeding sites in the Mutale (Bende Mutale), Nwanedi (Popallin Ranch) and Nzelele (Doreen Farms and Tshipise) rivers.

*Anopheles funestus* was historically a major vector of malaria in South Africa and continues to play a major role in malaria transmission in neighboring Mozambique and Zimbabwe. The threat of re-invasion by this species from neighboring countries cannot be ignored as evidenced by the very rapid and massive malaria resurgence following insecticide failure in KwaZulu-Natal Province in the late 1990′s [3,4]. Our surveys in 2017–2018, which yielded a solitary *An. funestus* female at Popallin Ranch close to the Zimbabwe border, suggests that any let-up in the annual indoor spraying programme may also create an opportunity for this species, as well as *An. gambiae s.s.*, to re-enter and establish a foothold in Limpopo.

### 4.3. Other Anophelines of Potential Vector Importance

During the initial years (or decades) of malaria vector control, it was resource-efficient to focus efforts on the known and most abundant vector species, which accounted for the overwhelming preponderance of transmission. However, as the target of malaria elimination approaches and residual malaria is confronted despite ongoing “traditional” vector control applications, it becomes important to consider the possible contribution of other mosquito species which could account for persistent low-incidence transmission, especially by species that feed outdoors and use cattle and other animals as additional blood-meal sources. Such “secondary” vectors may in fact then become the new “primary” vectors, and many such species have been incriminated in Africa [38].

Swellengrebel et al. [2] in 1931 reported parasite-infected *An. pretoriensis* in widely separate locations in South Africa. *Anopheles rufipes*, *An. coustani* and *An. squamosus* have all been found to host *P. falciparum* sporozoites elsewhere in Africa [38] and all these species were present in our surveys in the Limpopo River Valley, often commonly as with *An. rufipes* and *An. pretoriensis* (Figure 4).

The recent findings of *P. falciparum* infected *An. parensis* [48] and *An. vaneedeni* [55] in Mpumalanga and Kwazulu/Natal Provinces, require urgent investigations into the distribution of these species, their biology and their roles in transmission in other areas of the region. As the malaria elimination target is approached, more, not less, entomological investigations are needed to support the implementation of novel vector control strategies that will be required in order to reach the goal of elimination.

### 4.4. Importation of Infective Mosquitoes into South Africa

A proportion of local malaria cases in Limpopo Province are likely to have been caused by the importation of infective mosquitoes into South Africa from neighboring highly endemic regions, especially Mozambique. This phenomenon is known as odyssean malaria and is akin to the phenomenon of airport malaria except that it is postulated that most of the infective mosquitoes are inadvertently transported considerable distances by land. A system of migrant labour for the many gold and other mines, especially but not exclusively in Gauteng Province, depends on many thousands of Mozambicans and Zimbabweans coming in large measure from malaria high-burden rural areas in their respective countries. These people return home each year for annual vacation with their families and depend on bus transport. Such vehicles provide refuge for mosquitoes during overnight stops in those countries and bring infected *Anopheles* back into South Africa. A proxy for the frequency of such importation comes from the non-endemic Gauteng Province, in which 5–20 odyssean malaria cases are reported annually [60]. As it is not currently possible to distinguish between local and odyssean malaria in Limpopo Province, the actual frequency of this occurrence is unknown, but is a likely contributor to malaria epidemiology in all of South Africa’s malaria affected provinces.

## 5. Conclusions

A literature survey supports our field survey findings that *Anopheles arabiensis* is a rare species in the malaria-endemic Limpopo River Valley of Vhembe region, Limpopo Province, except for locations within the Kruger National Park wildlife preserve. This does not reflect sampling inadequacy, as the methods used were capable of detecting *An. arabiensis* in situations where it does occur, as in the Kruger National Park, by way of larval sampling, human landing catches and CO_2_-baited net or light traps.

Anopheline mosquitoes are not distributed homogeneously across the landscape either in space or time but vary in community composition/species richness in ways that are rarely easy to explain. Their abundance also varies across sites and over time again in ways that are not always predictable.

It is difficult to correlate patterns of malaria case incidence with presence and especially abundance of *An. arabiensis*, the species which is widely held to be the prime malaria vector in the drier inland savanna regions of southern Africa. It may be that other anopheline species are playing an as yet unrecognized role in malaria transmission, in particular members of the *An. funestus* group which are widespread and abundant in the region, but also other species such as *An. pretoriensis* and *An. rufipes*.

It should be recognized that known, highly effective vectors such as *An. funestus* and *An. gambiae s.s.*, remain prevalent in areas adjoining South Africa and the available evidence suggests that these species are kept at bay only because of continued widespread vector control interventions, in particular IRS. Although only one *An. funestus* and one *An. gambiae s.s.* were collected during our survey in the Limpopo River Valley, it demonstrates the potential for these species to re-colonize Limpopo Province if control efforts are reduced.

A small proportion of local malaria cases recorded in Limpopo Province are likely caused by the inadvertent importation of infective mosquitoes from neighbouring endemic regions.

Given that the available evidence suggests *An. arabiensis* is almost certainly not playing a significant role in malaria transmission in the Vhembe District, we propose that additional research effort, with greater and more frequent sampling intensity over a longer period, should be directed towards establishing which species are serving as vectors, so that appropriate control measures can be targeted at such species.

## Figures and Tables

**Figure 1 ijerph-17-04125-f001:**
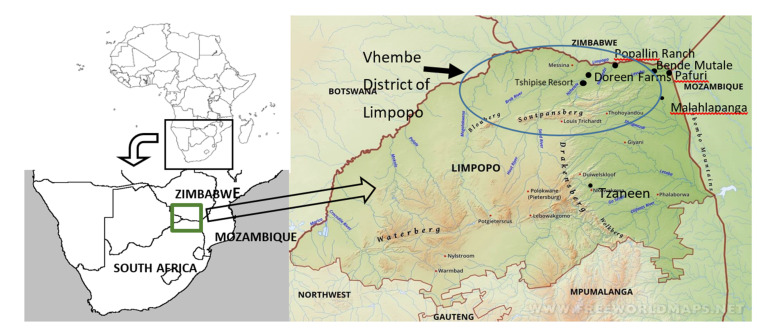
Limpopo River Valley study area.

**Figure 2 ijerph-17-04125-f002:**
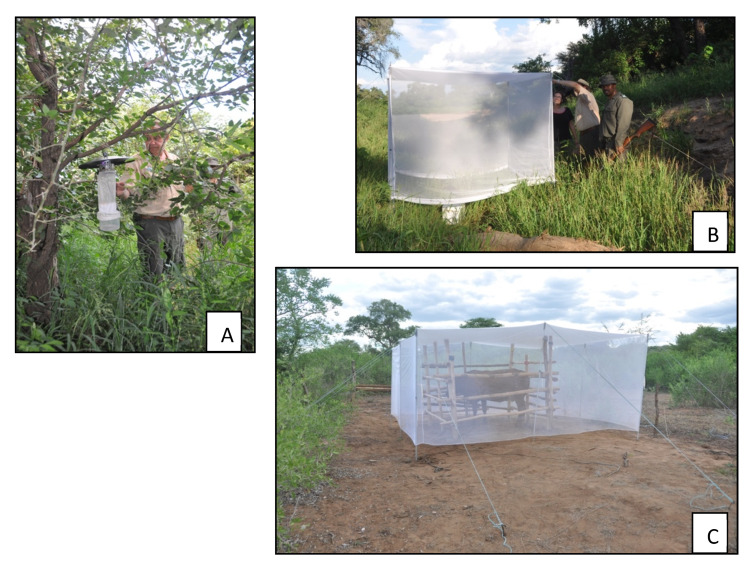
Trap types deployed during current study: (**A**)—CO_2_-baited CDC light trap; (**B**)—CO_2_-baited tent trap; (**C**)—cow-baited tent trap.

**Figure 3 ijerph-17-04125-f003:**
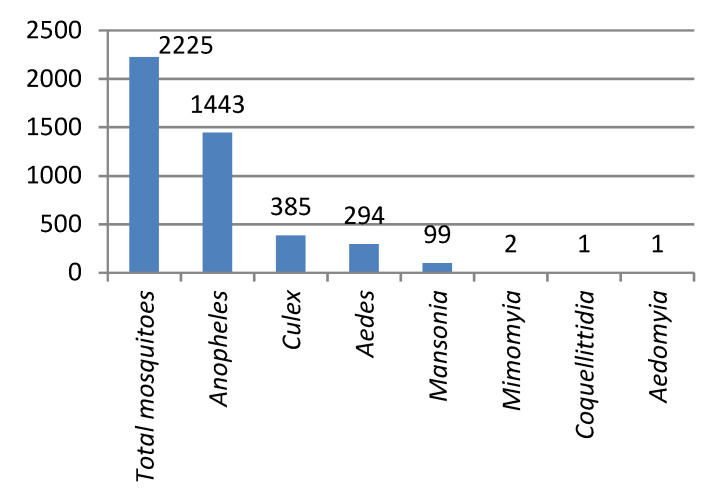
Mosquito numbers by genera: total adults collected October 2017–October 2018.

**Figure 4 ijerph-17-04125-f004:**
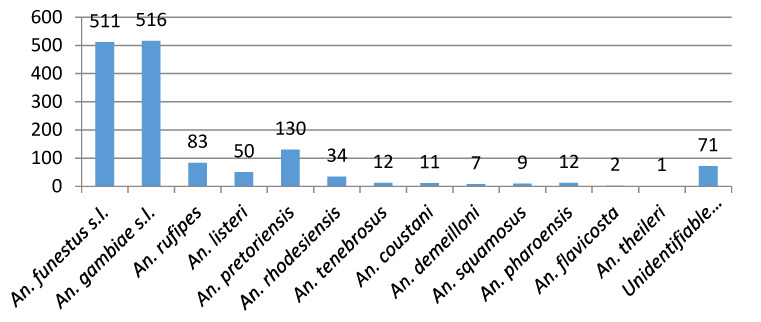
Mosquito numbers by *Anopheles* species: adults and larvae collected October 2017 to October 2018.

**Figure 5 ijerph-17-04125-f005:**
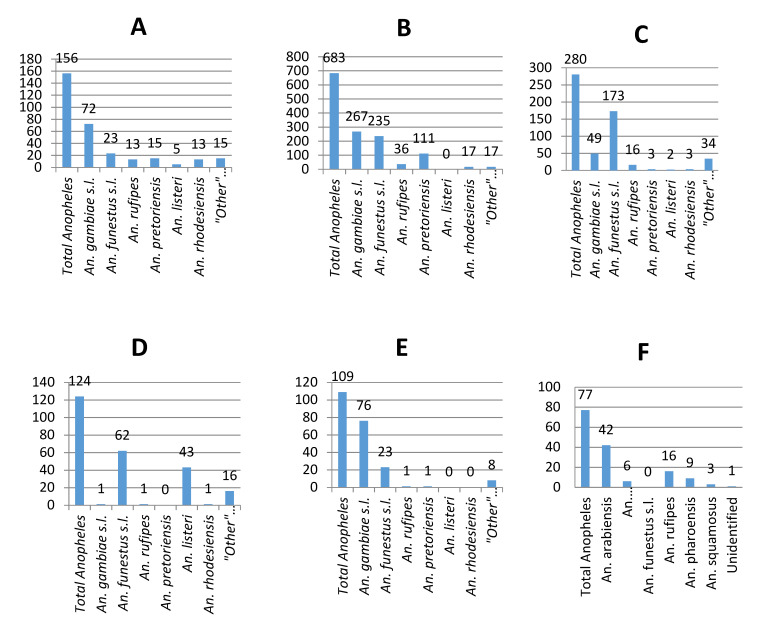
Adult *Anopheles* collected in 2017/18 at six localities: (**A**) Doreen Farms; (**B**) Bende Mutale; (**C**) Popallin Ranch; (**D**) Tshipise Resort; (**E**) Pafuri; (**F**) Malahlapanga.

**Figure 6 ijerph-17-04125-f006:**
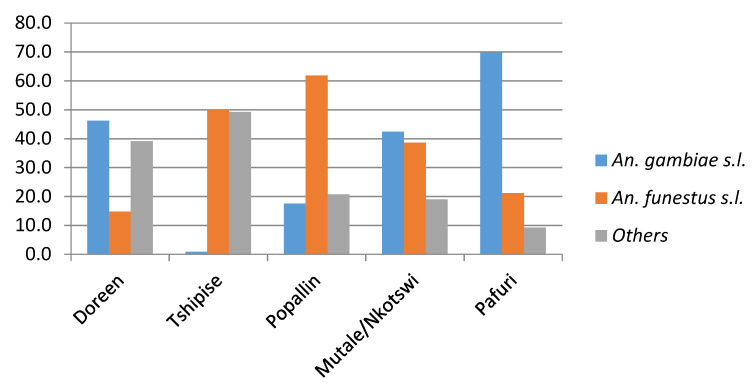
Percentage of *Anopheles funestus* group and *An. gambiae* complex collected in Limpopo Province in 2017/18.

**Table 1 ijerph-17-04125-t001:** Summary of key publications providing data on vectors for Limpopo Province, South Africa, and immediate neighboring regions.

Author(s) [Reference]	Date	Outline of Publication Content	Key Findings
Ingram & De Meillon [6]	1927	Mosquito survey in 1926, results covering northern Transvaal (the current Limpopo Province) and coastal Zululand (current KwaZulu-Natal), indicating distribution and breeding sites (larval collections having been the primary survey tool), with discussion around malaria vector species and control options.	Thirteen anopheline species/species groups found, speculating that *An. funestus* and *An. gambiae* are the main vectors despite strongly fluctuating population numbers and apparent extended absence even during malaria transmission periods.
Ingram & De Meillon [7]	1929	Mosquito survey in 1928, results covering northern and eastern Transvaal (Limpopo and Mpumalanga Provinces respectively) with discussion around malaria vector species and control options.	Thirteen anopheline species found, mostly through larval collections.
Swellengrebel et al. [2]	1931	Survey of anophelines in different habitat settings in “Transvaal” and “Zululand” to detect parasite positivity rates.	Six *Anopheles* species found indoors, and malaria parasites found in *An. funestus*, *An. gambiae* and also *An. pretoriensis*.
Steyn et al. [13]	1955	Two-week survey in March 1953 of mainly culicine mosquitoes by way of mostly larval collections, in the general area from Vaalwater to Musina in current Limpopo Province	538 mosquito specimens making up 21 species in three genera (*Anopheles* 6 species; *Aedes* 9 spp.; *Culex* 6 spp.)
La Grange & Coetzee [14]	1997	Anopheline survey 1987–1989 in Thomo Village, Limpopo Province, using human landing catches, outdoor resting catches, and larval rearing.	Exophilic members of *Anopheles funestus* group most abundant, comprising 85.8% of total *Anopheles* catch (*n* = 23,252). Of *Anopheles* landing on humans, *An coustani* was most abundant at 70.1% (*n* = 2994), followed by members of the *An. funestus* group at 28.1%. Of *An. gambiae* complex captured (*n* = 245) or reared (*n* = 225), 155 and 164 respectively were *An. quadriannulatus.* No *An. arabiensis* were found.
Govere et al. [15]	2000	Monthly collections of *Anopheles* at 7 sites in the Lowveld region of Mpumalanga Province, August 1997–May 1998, using human landing catches, window exit traps, and indoor knockdown spraying.	A total of 5084 *Anopheles* were collected, of which 2837 (55.8%) were *An. coustani*, 1418 (27.9%) *An. funestus* group, 435 (8.6%) *An. gambiae* complex, 264 (5.2%) *An. pretoriensis* and 130 (2.6%) a mix of other anopheline species. Of the *An. gambiae* complex, *An. merus* (56%) and *An. quadriannulatus* (30.4%) dominated, with *An. arabiensis* making up 13.6%.
Munhenga et al. [16]	2014	Anopheline species collected from five sites over two years in the northern Kruger National Park as part of an assessment of sites for possible sterile male release for malaria vector control.	A total of 3311 anophelines comprising nine species, showing clear and consistent differences in *Anopheles* community composition between sites even relatively close to each other.
Cornel et al. [17]	2018	Description of mosquito diversity and abundance at multiple sites across southern Africa, including Shingwedzi and Lapalala Nature Reserve in Limpopo Province.	Eight species of *Anopheles* comprising 63.1% of the total catch of 168 mosquitoes at Shingwedzi.

**Table 2 ijerph-17-04125-t002:** Survey of anopheline mosquitoes from the malaria endemic regions in the northern (Limpopo Province) and north-eastern (Mpumalanga Province) “Transvaal” [6,7]. P = present.

Species	Zoutpans-Berg 1926	Zoutpans-Berg 1928	Waterberg 1926	Waterberg 1928	Skukuza 1928	Tzaneen 1928	Musina 1928
*An. cinereus*			P	P		P	
*An. coustani*	P	P	P	P	P	P	
*An. demeilloni*	P			P		P	
*An. funestus* group	P	P		P	P	P	
*An. gambiae* complex	P	P		P	P	P	P
*An. longipalpis*	P	P		P		P	
*An. maculipalpis*	P						
*An. marshallii*				P		P	
*An. natalensis*	P					P	
*An. nili*					P		P
*An. pretoriensis*		P	P	P	P	P	P
*An. quadriannulatus*						P	
*An. rhodesiensis*		P	P	P	P		P
*An. rufipes*	P	P	P	P	P	P	P
*An. squamosus*		P	P	P		P	
*An. theileri*			P	P			

**Table 3 ijerph-17-04125-t003:** Parasite infections in *Anopheles* surveyed in the “Transvaal” in 1931 by Swellengrebel et al. [2].

Species	Letaba Foothills			Ofcolaco
	Inside Rural Huts in Foothills Number of Mosquitoes (Number Parasite-Infected)	Inside Rural Farmhouses Number of Mosquitoes (Number Parasite-Infected)	Outside Rural Farmhouses Number of Mosquitoes (Number Parasite-Infected)	Inside Rural Huts in Lowland Area Number of Mosquitoes (Number Parasite-Infected)
*An. funestus* group	240 (44)	44 (6)	7 (0)	53 (0)
*An. gambiae* complex	6 (0)	-	1 (0)	161 (27)
*An. maculipalpis*	-	1 (0)	-	-
*An. marshallii*	-	-	-	1 (0)
*An. pretoriensis*	9 (0)	1 (1)	110 (1)	-
*An. rufipes*	4 (0)	-	14 (0)	1 (0)

**Table 4 ijerph-17-04125-t004:** Findings of a 1953 survey of mosquitoes in the upper Limpopo River Valley [13].

Species	Larvae	Adults	Total
*Anopheles coustani*	7	2	9
*An. gambiae* complex	23	3	26
*An. listeri*	19	-	19
*An. rufipes*	2	1	3
*An. pretoriensis*	12	1	13
*An. squamosus*	-	1	1
*Aedes* spp. *(scatophagoides*, *fulgens*, *aegypti*, *metallicus*, *calceatus*, *vittatus*, *marshalli*, *dentatus*, *hirsutus)*	257	71	328
*Culex* spp. *(tigripes*, *nebulosus* var. *pseudocinereus*, *theileri*, *univittatus*, *simpsoni*, *decens)*	88	51	139
Totals	408	130	538

**Table 5 ijerph-17-04125-t005:** *Anopheles* mosquitoes collected from Thomo village, Limpopo Province, from June 1987 to August 1989 [14]. F = females; M = males; HLC = human landing catches; Pit = pit collections; Natural = natural refuges.

Species	HLC	Pit F	Pit M	Natural F	Natural M	Cattle Enclosures	Total
*An. funestus* group	842	9234	6807	1819	1099	157	19,958
*An. gambiae* complex	6	45	20	115	51	8	245
*An. coustani*	2100	3	3	9	1	21	2137
*An. rufipes*	6	41	22	343	361	24	797
*An. squamosus*	25	-	-	1	-	6	32
*An. pretoriensis*	-	-	1	17	11	-	29
*An. marshallii*	2	2	-	5	-	1	10
*An. pharoensis*	4	-	-	-	-	3	7
*An. longipalpis*	5	4	-	3	-	1	13
*An. demeilloni*	1	2	-	2	-	3	8
*An. maculipalpis*	2	-	-	3	-	9	14
*An. theileri*	1	-	-	-	-	1	2
Totals	2994	9331	6853	2317	1523	234	23,252

**Table 6 ijerph-17-04125-t006:** *Anopheles* mosquitoes collected from five sites in the northern Kruger National Park, Limpopo Province, July 2010 to December 2012 [16].

Species	Total Collected	Percentage Composition (Aggregate of Specimens Caught at All Sites)	Number of Sites Collected from
*An. arabiensis*	1352 **	44.3	3
*An. quadriannulatus*	870	28.5	4
*An. merus*	349 ***	11.4	2
*An. coustani*	395	12.9	3
*An. pretoriensis*	35	1.1	2
*An. maculipalpis*	28	0.9	3
*An. rivulorum*	19	0.6	2
*An. squamosus*	3	0.1	1
*An. rufipes*	2	0.1	1

** of which 99.6% collected at one site, Malahlapanga freshwater spring. *** of which 98.9% collected at the two salt-water springs Mafayeni & Matiovila.

**Table 7 ijerph-17-04125-t007:** *Anopheles* species collected by CO_2_-baited net and light traps at Shingwedzi, Kruger National Park and Lapalala Nature Reserve, Limpopo Province in 2015 [17].

Locality	*Anopheles* (n)	No. *Culex*	No. Other Genera	Summary
Shingwedzi River, Kruger National Park, eastern Limpopo Province	*An. leesoni* (2) *An. parensis* (1) *An. pharoensis* (3) *An. quadriannulatus* (66) *An. rivulorum* (6) *An. rivulorum*-like (26) *An. squamosus* (4) *An. theileri* (1) Total: 109	25	34	109 *Anopheles* out of 168 mosquitoes = 65%
Lapalala Nature Reserve, western Limpopo Province	*An. coustani* (41) *An. longipalpis* (1) *An. marshallii* (54) *An. squamosus* (6) *An. theileri* (42) Total: 144	43	109	144 *Anopheles* out of 296 = 49%

**Table 8 ijerph-17-04125-t008:** Monthly human landing captures off eight humans sitting 18:00 to 22:00, usually four nights per month at seven sites in Mpumalanga Province, August 1997 to May 1998 [15].

Month	Sampling Days (%)	*An. coustani*	*An. funestus* Group	*An. gambiae* Complex	*An. pretoriensis*	Other *Anopheles*
August 1997	15 (12.0)	809	579	6	20	9
September	17 (13.6)	650	321	8	9	1
October	20 (16.0)	518	227	63	5	18
November	17 (13.6)	291	212	95	14	7
December	8 (6.4)	142	0	53	35	15
January 1998	14 (11.2)	105	12	117	17	6
February	15 (12.0)	251	23	77	60	13
March	7 (5.6)	55	6	3	44	7
April	4 (3.2)	14	6	4	39	14
May	8 (6.4)	2	32	9	21	40
Total (%)	125 (100.0)	2837 (55.8%)	1418 (27.9%)	435 (8.6%)	264 (5.2%)	130 (2.6%)

**Table 9 ijerph-17-04125-t009:** Species composition (%) and abundance of anophelines captured by sampling method and locality in eastern Zimbabwe [28].

Sampling Method	Sampling Region	Total *Anopheles*	*An. funestus* Group	*An. gambiae* Complex	*An. pretoriensis*
Pyrethrum spray catch	Burma Valley Ward, Mutare District	795	96.6%	3.3%	0.1%
	Zindi Ward, Mutasa District	140	96.4%	3.6%	0%
Reared from larvae	Burma Valley Ward, Mutare District	3141	1.4%	0.2%	98.4%
	Zindi Ward, Mutasa District	1549	2.9%	0.2%	96.9%
Totals		5625	17.5%	0.8%	81.7%

**Table 10 ijerph-17-04125-t010:** Polymerase Chain Reaction-identified *Anopheles* species collected in the northern Limpopo Province, South Africa.

Locality	Sampling Month	*An. funestus*	*An. leesoni*	*An. parensis*	*An. rivulorum*	*An. rivulorum-*Like	*An. vaneedeni*	*An. gambiae s.s.*	*An. arabiensis*	*An. quadriannulatus*	*An. merus*
Bende Mutale	Feb 2018	-	3	-	165	3	-	-	-	-	-
Bende Mutale	Apr 2018	-	2	-	2	9	-	-	1	59	6
Bende Mutale	Oct 2018	-	7	2	7	-	-	-	-	35	1
Popallin Ranch	Oct 2017	-	2	-	47	-	-	-	-	-	-
Popallin Ranch	Feb 2018	1	1	2	29	-	1	1	-	-	-
Popallin Ranch	Apr 2018	-	2	-	54	7	-	-	-	4	1
Doreen Farms	Feb 2018	-	-	-	5	-	2	-	-	-	-
Doreen Farms	Apr 2018	-	4	-	2	-	1	-	1	10	-
Tshipise Resort	Apr 2018	-	4	-	1	1	42	-	1	-	-
Nkotswi	Apr 2018	-	-	-	-	-	-	-	-	7	-
Tshikuyu	Apr 2018	-	-	-	-	-	-	-	-	1	-
Pafuri	Apr 2018	-	-	-	-	-	-	-	3	27	-
Malahla-panga	Oct 2018	-	-	-	-	-	-	-	42	7	-
Total		1	25	4	312	20	46	1	48	150	8

## Data Availability

The data supporting the conclusions of this article are included within the article. Enquiries for more detailed data should be directed to the Corresponding Author.

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
