# Peer review of "Malaria Vectors and Vector Surveillance in Limpopo Province (South Africa): 1927 to 2018"

_ijerph, 2020, doi:10.3390/ijerph17114125_

Round 1

Reviewer 1 Report

I would like to commend the authors on this excellent extensive review and really well written. I don't have any major comments, only 3 minor comments.

  1. title 2.6 is a bit too long, maybe " Vector Surveillance and Methodology"
  2. in conclusion, L469-470, "demonstrates the potential for these species to rapidly re-colonize" while you captured only one mosquito of each species, sounds a bit too excessive. I suggest rephrasing this sentence.
  3. the conclusion ends a bit abruptly. Something is missing such as the future plans, perspectives, or implications of your findings.

Author Response

RESPONSE TO COMMENTS BY REVIEWER 1.

Comments and Suggestions for Authors

I would like to commend the authors on this excellent extensive review and really well written. I don't have any major comments, only 3 minor comments.

  1. title 2.6 is a bit too long, maybe " Vector Surveillance and Methodology" Response: We have shortened the title as suggested.
  2. in conclusion, L469-470, "demonstrates the potential for these species to rapidly re-colonize" while you captured only one mosquito of each species, sounds a bit too excessive. I suggest rephrasing this sentence. Response: We have deleted the “rapidly”. However, we would nevertheless like to point out that extremely rapid re-colonizations has indeed happened once before in South Africa when a change in IRS insecticide occurred in KwaZulu Province and Anopheles funestus re-entered from Mozambique, with massive rise in malaria cases, as we wrote in Lines 413 and 414 in the line numbering of the previous manuscript. So “rapid re-colonize” is not necessarily an exaggeration, but we defer to the Reviewer.
  3. the conclusion ends a bit abruptly. Something is missing such as the future plans, perspectives, or implications of your findings. Response: We have added an appropriate ending to round off the Conclusion more meaningfully.

Reviewer 2 Report

This MS reviews the main malaria vectors in the northeast of South Africa (Limpopo Province). To that end, the authors conducted a review of the published literature about these vectors between 1927 and 2018, and compare them with the data of their own recent campaign (2017-2018). From these data, they conclude that Anopheles arabiensis is a rare species in the valley of the river Limpopo, unlike in other areas of South Africa.

Rather than a review, this is an update of the situation of these vectors, comparing the results obtained in a recent sampling with those of the existing historic studies. It is potentially a study of great interest to be published. Some aspects are valuable, such as the Results and Conclusions sections. However, it requires some major changes and minor corrections to be accepted for publication.

General remarks:

. Title. We recommend the authors to better adjust the title to the content. It is not an extensive bibliographic review, but an update. We suggest that, instead of “Review of Malaria vectors and Vector Surveillance…”, you should consider something in the line of “Historical overview of Malaria vectors and Vector Surveillance…”. It would also be interesting to include the temporal period of the review in the title (20th and 21st centuries).

. There are several sections that require an important improvement in order to be published: Abstract, Study Area and Figures 1-3.

. Abstract. In its current version it is confusing and disordered. The Abstract begins directly (p. 1 Lines 17-22) with the Results of the sampling of 2017-2018 and the Methodology. It must begin with the clarification of the research interest and object of study (briefly including the Study Area). From there, the authors can include the Methodology and Results. The results presented in the Abstract are not in the same order as in the MS. We recommend you to present them as they are in the MS: first a brief synthesis of the results of the historical samplings and then a comparison of these with the results of your survey of mosquitoes of 2017-2018. Finally, you can add a part of the Discussion and Conclusions.

. Introduction and Objectives. These are not in line with the results and conclusions. The main conclusions of this MS are focused on demonstrating that Anopheles arabiensis is a rare species in the valley of the river Limpopo, unlike in other areas of South Africa. Likewise, the obtained results show the role that other members of the An. funestus group, especially An. funestus and An. gambiae s.s., can play as malaria vectors. However, An. arabiensis and A. funestus are not mentioned in the Introduction. Moreover, the objectives of the MS (p. 2 lines 60-62) are somewhat confusing, but they can be easily improved: this study is a historical review of the published works about the main vectors that transmit malaria in the northeast of South Africa (Limpopo Province). To this end, the authors carried out a review of the published literature about these vectors from 1927 to 2018, and updated the obtained information with the data of their own recent campaign conducted in 2017-2018. Please, adapt or re-write the objectives considering these aspects.

. Study Area and Figures 1-3. This section must be thoroughly revised. We recommend ordering the text and using original figures that are consistent with the text in the MS. Figures 1-3 are hard to read (especially Figures 2 and 3). Some of them are illegible (e.g., the small map in the bottom left corner of Figure 3) or include too much text (Figure 2). Figures should not have so much text as in the case of Figure 2; besides, a large part of that text is irrelevant (e.g., the Game Reserve Key list in Figure 2). Please, make sure you have permission to publish these figures (e.g., Figure 2 has an institutional stamp and Figure 3 has a water mark).

Instead of the current figures, which are confusing, this section should gather, first of all, the location (at the African regional scale) of the study area (including South Africa and all its provinces), the delimitation of the province of Limpopo and the Kruger National Park in more detail, as well as the sampling points of 2017-2018 (including the place names that are cited in the text and in the figures).

Furthermore, the graphic representation of the historic distribution of malaria in South Africa in 1938 (Figure 1) and nowadays (Figure 2) can be of great interest if you represent them jointly and in comparable formats. Therefore, we recommend you to remake these two figures to represent them together (and more originally) or remove them. Figure 1 (p. 2) better illustrates section “3.1. Historical Surveys in Limpopo Province” around p. 7. In its current presentation (Study Area), the historic representation of 1938 by Coetzee et al. (2013 p. 772) cannot be understood. Please, make sure that you have permission to reproduce this figure.

Please, reorder also the text of section 2.2. In its current presentation, it is confusing, too brief and difficult to follow.

Specific remarks:

. P. 1 line 33. This sentence is incomplete. What is the current global situation of endemic malaria in South Africa? You can complete this with the information of lines 42-44 (p. 1).

. P. 2 Line 65. Replace "The geographic focus of this paper..." with "The study area is located..." or something similar.

. P. 2 Lines 67-69. Add bibliographic citation(s) to support this information.

. P 4 Data sources. Lines 90-93. What criteria and keywords did you use in your bibliographic search? Did you use geographic criteria in your search? What time period does it comprise? From 1927 to 2018? These aspects must be detailed as much as possible.

. P. 6 Figure 5. The presented image is not good enough to be published. Improve the quality of the image or remove it (it does not seem essential for understanding section 3.1).

. P. 13 line 299. In botany, whenever a taxon is mentioned for the first time in an MS, the name of the author who first described it must be included. Write Phragmites australis (Cav.) Trin. ex Steud., instead of Phragmites australis. http://www.theplantlist.org/tpl1.1/record/kew-433921

. Line 161, Table 4. Remove the italics in “spp” and add full stop, that is, “spp.”.

. Fig. 9. It does not seem appropriate to add a title inside Figure 9. Please, put the names of the species in italics and “s.l.” instead of “s.l”. The same goes for Figures 6-9. In general, all figures are better edited without the frame. Carefully revise all these aspects.

. Revise the page numbering. From the section change (p. 14), the page numeration starts anew with page 1 instead of page 15.

. One of the most interesting conclusions of this MS is the potential role of the bordering regions around the study area (with the participation of other anophelines). Moreover, the involuntary importation of mosquitoes from neighbour endemic regions seems probable. Can you explain how mosquitoes are transported to far distances from the bordering areas such as Mozambique (lines 440-442 and 471-472)? In this MS, you discard means of transportation associated with airplanes (airport malaria) and mention the presence of seasonal workers. How do these workers transmit anophelines? Can you further expand section 4.4?

. In the case of A. arabiensis, there does not seem to be a simple cause that prevents it from appearing in the Limpopo province. This conclusion is also interesting (lines 450-452) Which could those causes be? This species is usually favoured in dry savannah environments and sparse woodlands with recent land disturbance, and it is both zoophilic and anthropophilic. Can it be affected by the presence of wild animals and livestock or woodland ecosystems with little disturbance such as (KNP)?

Author Response

RESPONSE TO COMMENTS BY REVIEWER 2.

Comments and Suggestions for Authors

This MS reviews the main malaria vectors in the northeast of South Africa (Limpopo Province). To that end, the authors conducted a review of the published literature about these vectors between 1927 and 2018, and compare them with the data of their own recent campaign (2017-2018). From these data, they conclude that Anopheles arabiensis is a rare species in the valley of the river Limpopo, unlike in other areas of South Africa.

Rather than a review, this is an update of the situation of these vectors, comparing the results obtained in a recent sampling with those of the existing historic studies. It is potentially a study of great interest to be published. Some aspects are valuable, such as the Results and Conclusions sections. However, it requires some major changes and minor corrections to be accepted for publication.

General remarks:

. Title. We recommend the authors to better adjust the title to the content. It is not an extensive bibliographic review Response: We respectfully disagree. This paper does in fact contain an exhaustive review of ALL publications EVER published on vectors in Limpopo Province. Nevertheless, we have removed “Review of…” from the title, but an update. We suggest that, instead of “Review of Malaria vectors and Vector Surveillance…”, you should consider something in the line of “Historical overview of Malaria vectors and Vector Surveillance…Response: To accommodate the concerns of both Reviewers, we have deleted the reference to “Review of…” in order to shorten the title (as recommended by Reviewer 1) and to respond to issues raised by Reviewer 2. It would also be interesting to include the temporal period of the review in the title (20th and 21st centuries) Response: we have added the dates.

. There are several sections that require an important improvement in order to be published: Abstract, Study Area and Figures 1-3. Response: We have amended all these sections in accordance with Reviewer 2 suggestions as indicated in various places in his/her comments elsewhere.

. Abstract. In its current version it is confusing and disordered. The Abstract begins directly (p. 1 Lines 17-22) with the Results of the sampling of 2017-2018 and the Methodology. It must begin with the clarification of the research interest and object of study (briefly including the Study Area). From there, the authors can include the Methodology and Results. The results presented in the Abstract are not in the same order as in the MS. We recommend you to present them as they are in the MS: first a brief synthesis of the results of the historical samplings and then a comparison of these with the results of your survey of mosquitoes of 2017-2018. Finally, you can add a part of the Discussion and Conclusions. Response: we have amended the Abstract accordingly.

. Introduction and Objectives. These are not in line with the results and conclusions. The main conclusions of this MS are focused on demonstrating that Anopheles arabiensis is a rare species in the valley of the river Limpopo, unlike in other areas of South Africa. Likewise, the obtained results show the role that other members of the An. funestus group, especially An. funestus and An. gambiae s.s., can play as malaria vectors. However, An. arabiensis and A. funestus are not mentioned in the Introduction Response: We have now inserted appropriate wording in the Introduction. Moreover, the objectives of the MS (p. 2 lines 60-62) are somewhat confusing, but they can be easily improved: this study is a historical review of the published works about the main vectors that transmit malaria in the northeast of South Africa (Limpopo Province). To this end, the authors carried out a review of the published literature about these vectors from 1927 to 2018, and updated the obtained information with the data of their own recent campaign conducted in 2017-2018. Please, adapt or re-write the objectives considering these aspects Response: We have amended the wording accordingly.

. Study Area and Figures 1-3. This section must be thoroughly revised. We recommend ordering the text and using original figures that are consistent with the text in the MS. Figures 1-3 are hard to read (especially Figures 2 and 3). Some of them are illegible (e.g., the small map in the bottom left corner of Figure 3) or include too much text (Figure 2). Figures should not have so much text as in the case of Figure 2; besides, a large part of that text is irrelevant (e.g., the Game Reserve Key list in Figure 2). Please, make sure you have permission to publish these figures (e.g., Figure 2 has an institutional stamp and Figure 3 has a water mark). Response: We have deleted Figures 1,2 and 5 as under Covid-19 lockdown conditions we do not have the capacity to develop new maps to standardize or integrate, for example, Maps 1 and 2. This deletion of Figures 1, 2 and 5 does not affect interpretation or other intrinsic value of the manuscript. We have re-developed the map of the study area in line with what the Reviewer wanted, and it is now labelled as Figure 1.

Instead of the current figures, which are confusing, this section should gather, first of all, the location (at the African regional scale) of the study area (including South Africa and all its provinces), the delimitation of the province of Limpopo and the Kruger National Park in more detail, as well as the sampling points of 2017-2018 (including the place names that are cited in the text and in the figures). Response: We have done this, and it is labelled as Figure 1 now that the other maps have been deleted.

Furthermore, the graphic representation of the historic distribution of malaria in South Africa in 1938 (Figure 1) and nowadays (Figure 2) can be of great interest if you represent them jointly and in comparable formats. Therefore, we recommend you to remake these two figures to represent them together (and more originally) or remove them. Figure 1 (p. 2) better illustrates section “3.1. Historical Surveys in Limpopo Province” around p. 7. In its current presentation (Study Area), the historic representation of 1938 by Coetzee et al. (2013 p. 772) cannot be understood. Please, make sure that you have permission to reproduce this figure. Response: We have deleted Figures 1, 2 and 5.

Please, reorder also the text of section 2.2. In its current presentation, it is confusing, too brief and difficult to follow Response: We have changed and added to the wording in section 2.2. as requested.

Specific remarks:

. P. 1 line 33. This sentence is incomplete. What is the current global situation of endemic malaria in South Africa? You can complete this with the information of lines 42-44 (p. 1). Response: Done.

. P. 2 Line 65. Replace "The geographic focus of this paper..." with "The study area is located..." or something similar. Response: Done.

. P. 2 Lines 67-69. Add bibliographic citation(s) to support this information. Response: Done.

. P 4 Data sources. Lines 90-93. What criteria and keywords did you use in your bibliographic search? Did you use geographic criteria in your search? What time period does it comprise? From 1927 to 2018? These aspects must be detailed as much as possible. Response: We have amended the wording to capture these requirements.

. P. 6 Figure 5. The presented image is not good enough to be published. Improve the quality of the image or remove it (it does not seem essential for understanding section 3.1). Response: Figure has been removed.

. P. 13 line 299. In botany, whenever a taxon is mentioned for the first time in an MS, the name of the author who first described it must be included. Write Phragmites australis (Cav.) Trin. ex Steud., instead of Phragmites australis. http://www.theplantlist.org/tpl1.1/record/kew-433921 Response: We have corrected as requested, thank you.

. Line 161, Table 4. Remove the italics in “spp” and add full stop, that is, “spp.”. Response: Corrections made as requested.

. Fig. 9. It does not seem appropriate to add a title inside Figure 9. Please, put the names of the species in italics and “s.l.” instead of “s.l”. The same goes for Figures 6-9. In general, all figures are better edited without the frame. Carefully revise all these aspects. Response: We have done as requested, except for the new Figure 5 where we have left the inside headings as there are multiple graphs and it is easier for the reader to understand which geographic area the data represents.

. Revise the page numbering. From the section change (p. 14), the page numeration starts anew with page 1 instead of page 15. Response: The Page Numbering was inserted by the Journal Editorial team, and we will request them to fix it before publication as we are unable to fix this re-formatted version.

. One of the most interesting conclusions of this MS is the potential role of the bordering regions around the study area (with the participation of other anophelines). Moreover, the involuntary importation of mosquitoes from neighbour endemic regions seems probable. Can you explain how mosquitoes are transported to far distances from the bordering areas such as Mozambique (lines 440-442 and 471-472)? In this MS, you discard means of transportation associated with airplanes (airport malaria) and mention the presence of seasonal workers. How do these workers transmit anophelines? Can you further expand section 4.4? Response: We have added wording to explain how this happens.

. In the case of A. arabiensis, there does not seem to be a simple cause that prevents it from appearing in the Limpopo province. This conclusion is also interesting (lines 450-452) Which could those causes be? This species is usually favoured in dry savannah environments and sparse woodlands with recent land disturbance, and it is both zoophilic and anthropophilic. Can it be affected by the presence of wild animals and livestock or woodland ecosystems with little disturbance such as (KNP)? Response: We are unable to provide any reasonable explanation for this very low abundance of Anopheles arabiensis in Limpopo Province, and imply it in our statement “Such shifts highlight our poor understanding of the factors affecting population distribution and abundance” in section 4.1. We are reluctant to offer speculative possible explanations.

Round 2

Reviewer 2 Report

The MS has improved significantly. In the current version it is much more attractive to read. The solutions to figures 1, 2 and 5 of the original version are correct. Just a minor comment:

The little map in the lower left corner of Figure 1 (P. 3 lines109-112) cannot be seen easily. We recommend that you improve its contrast and/or size.

Author Response

We are grateful to you for your attention to detail and acknowledge that your inputs have significantly improved the manuscript, many thanks for that. We have now changed Figure 1 (the map) to accommodate your suggestion, which was a good suggestion. All best wishes. Leo Braack